# Mutant *lpa1* Analysis of *ZmLPA1* Gene Regulates Maize Leaf-Angle Development through the Auxin Pathway

**DOI:** 10.3390/ijms23094886

**Published:** 2022-04-28

**Authors:** Xiangzhuo Ji, Qiaohong Gao, Fenqi Chen, Mingxing Bai, Zelong Zhuang, Yunling Peng

**Affiliations:** 1College of Agronomy, Gansu Agricultural University, Lanzhou 730070, China; 18693000306@163.com (X.J.); gaoqh3324@163.com (Q.G.); 18893115958@163.com (F.C.); baimingxing12@163.com (M.B.); zhuangzl3314@163.com (Z.Z.); 2Gansu Provincial Key Laboratory of Aridland Crop Science, Gansu Agricultural University, Lanzhou 730070, China; 3Gansu Key Laboratory of Crop Improvement & Germplasm Enhancement, Lanzhou 730070, China

**Keywords:** maize (*Zea mays* L.), *ZmLPA1*, leaf angle, IAA, auxin pathway

## Abstract

Maize plant type is one of the main factors determining maize yield, and leaf angle is an important aspect of plant type. The rice Loose Plant Architecture1 (*LPA1*) gene and Arabidopsis AtIDD15/SHOOT GRAVITROPISM5 (*SGR5*) gene are related to their leaf angle. However, the homologous ZmLPA1 in maize has not been studied. In this study, the changing of leaf angle, as well as gene expression in leaves in maize mutant *lpa1* and wild-type ‘B73’ under different IAA concentrations were investigated. The regulation effect of IAA on the leaf angle of *lpa1* was significantly stronger than that of the wild type. Transcriptome analysis showed that different exogenous IAA treatments had a common enrichment pathway—the indole alkaloid biosynthesis pathway—and among the differentially expressed genes, four genes—AUX1, AUX/IAA, ARF and SAUR—were significantly upregulated. This study revealed the regulation mechanism of *ZmLPA1* gene on maize leaf angle and provided a promising gene resource for maize breeding.

## 1. Introduction

Maize (*Zea mays* L.) is the most productive crop in the world. In addition to being used for food, maize is mainly used as feed and for industrial raw materials, and it occupies a pivotal position in the world economy [1]. High yield has always been the goal of crop-industry development, and plant type is one of the important determinants in maize yield [2,3]. Leaf angle is a crucial aspect of plant type, and analysis of the cloning and molecular mechanism of leaf-angle-related genes is the basis for the ideal plant-type breeding of maize [4,5,6,7]. Different plant types affect the light distribution and light-energy utilization in the canopy of the population, and this ultimately affects the yield [7,8]. A complete maize leaf consists of three parts: the leaf blade, leaf pillow (including leaf tongue and leaf ear) and leaf sheath. The acute angle between the leaf and the main stem is the leaf angle. According to the difference in leaf angle, maize plant types are divided into three types, namely compact type, intermediate type and flat type [9]. The leaf angle is one of the main morphological indicators to measure the compactness of maize [8,10]. There is a correlation between the maize leaf angle and yield: the smaller the upper leaf angle, the lower the light-interception ability, which is beneficial to the middle leaf for receiving light, improving the production and accumulation of dry matter, and then increasing the yield [11].

Previous studies found that leaf-angle development is jointly regulated by a variety of plant hormones, and auxin plays a crucial role in regulating leaf-angle formation [12]. Indole-3-acetic acid (IAA) is the main form of auxin in maize, which mainly regulates apical dominance, root heaviness, promotion of cell elongation, inflorescence formation and embryo development [13]. Studies have found that an appropriate level of auxin plays an important role in the normal development of leaf primordium (leaf primordium) and the formation of leaves. The polar transport of auxin triggers the formation of shoot apical meristem (SAM) leaves [14]. In maize, inhibiting the transport of auxin will destroy the production of new leaves, affecting the growth of existing leaves and the formation of leaf edges. Guxin-Amino Synthetase 3 (GH3) is an early auxin-responsive gene widely present in plants [15]. In rice, *OsGH* participates in the microRNA-mediated auxin signal-transduction pathway and maintains the homeostasis of auxin [16,17]. In the rice mutant leaf inclination1 (*lc1-D*), the content of free auxin is reduced, showing a phenotype of increased leaf angle [18]. Rice *LAZY1* regulates the polar transport of auxin. Corresponding rice and Arabidopsis *lazy* mutants both exhibit a weakened aerial gravitational response and an increase in tiller and leaf angles [19,20,21].

Ku et al. used inbred lines Yu 82 and Shen 137 to construct an F2:3 family and located the *ZmTAC1* gene that regulates leaf angles in the qLA2 region of the second chromosome by the CAP marker method. The nucleic acid sequence of the two parents is different at the 5′-UTR end, and its ‘CTCC’ becomes ‘CCCC’, which affects the expression level of *ZmTAC1*, thereby further affecting the size of the leaf angle [22]. Juarez et al. found that mutants *rld1* and *lbl1* showed paraxial/upward leaf morphology. By cloning the genes corresponding to these two mutants, it was found that *rld1* encodes a HD-ZIPIII protein, and the semidominant Rldl-O mutant caused continuous expression of the distal transcript due to a base substitution at the miR166 complementary site, causing the blade to deflect to the proximal end [23]. Zhang et al. localized the *ZmCLA4*, which controls leaf angle, to the qLA4-1 region by fine positioning. *ZmCLA4* is a homologous gene of *LAZY1* in rice and Arabidopsis; the accumulation of *ZmCLA4* mRNA is negatively correlated with the change in leaf angle, and then negatively regulates the development of the leaf angle [24]. Josh et al. found that *drl1* mutants exhibited leaf-drooping morphology, and *drl1* mutations affected leaf length, width and leaf angle [25].

At present, *LPA1* has been reported to regulate leaf angle in rice; it specifically regulates the gravitational response of rice, which increases the tillering angle and leaf angle, resulting in a loose rice plant. *LPA1* is the functional ortholog of AtIDD15/SGR5, but with distinct features [26,27]. However, *ZmLPA1* as a homologous gene of *LPA1* has not been reported in maize, and the mechanism of regulating plant type has not been studied. In this paper, we obtained the *ZmLPA1* mutant *lpa1* by EMS mutagenesis of wild-type B73, which displays a relatively larger leaf angle, maize mutant *lpa1* and wild-type ‘B73’ were analyzed using RNA-Seq sequencing, aiming at clarifying the molecular mechanism of the *ZmLPA1* gene regulation leaf angle in maize.

## 2. Results

### 2.1. Identification of ZmLPA1 Gene

The *ZmLPA1* gene sequence (LOC103633308) was obtained from the NCBI database. The gene is located on maize chromosome 1. The full-length gene sequence is 2120 bp, including a 5′-UTR of 375 bp and a 3′-UTR of 387 bp. The cDNA sequence is 1359bp, encoding 452 amino acids. By comparing the DNA and cDNA sequences, it was found that the full length of the gene consists of three exons and two introns. The mutation position of the mutant *lpa1* occurred at the 691st base of the *ZmLPA1* gene sequence, and a T-C base was replaced to mutate cysteine (Cys) to arginine (Arg), Mutations lpa1 in the ZmLPA1 gene lead to changes in leaf angle in wild-type B73 maize. (Figure 1A). The protein sequence of ZmLPA1 was analyzed by Blastp in GeneBank, and it was found that it has high homology with the protein sequences of *Panicum hallii*, *Panicum miliaceum*, *Dichanthelium oligosanthes*, *Setaria italica*, *Sorghum bicolor*, *Hordeum vulgare*, A*egilops tauschii*, *Brachypodium distachyon*, *Oryza sativa*, *Glycine max*, *Phoenix dactylifera* and *Elaeis guineensis*. The above plant-protein sequences were compared by MEGA7.0 software, and then a phylogenetic tree was constructed. The results showed that a variety of ZmLPA1 homologous protein sequences were found, which were widely distributed in angiosperms, indicating that the ZmLPA1 may originate in angiosperms. After the emergence of plants, it has specific effects on higher plants. The ZmLPA1 was divided into two branches between dicotyledonous and monocotyledonous plants, and it could be seen that ZmLPA1 was most closely related to maize (GRMZM2G074032) and sorghum (XP_021312760.1) (Figure 1B). To clarify the expression specificity of the *ZmLPA1* in different maize tissues, the expression patterns of the *ZmLPA1* in maize root, stem, leaf, ear and tassel were detected by qPCR technology. The results showed that the expression patterns of the *ZmLPA1* in maize tissues were different. The highest expression level was in maize roots, which was more than 5 times that of other tissues and organs. There may be a close relationship between the *ZmLPA1* and auxin in the process of maize growth and development (Figure 1C).

### 2.2. Phenotype Identification of Maize Mutant lpa1

The phenotypic analysis of the mutant *lpa1* and its wild-type B73 showed that the mutant has phenotypic variation in various organs (Figure 2 and Figure 3). On the whole, the whole plant type of the mutant is quite different from the wild type. Compared with the wild type, the height of the mutant is significantly lower than that of the wild type. From the trumpet stage, the leaf angle is significantly different; the leaf angle of the mutant is larger than that of the wild type. When the leaf angle of the mutant is 45.3° while that of the wild type is only 30.1°, the main stem and leaf ears of the mutant are red, the number of tassel branches of the mutant is significantly reduced, and the pollen vigor is weak. In addition, the mature ears of the mutant become smaller, and the grains are oblate and smaller. When the maize grows to the pollination period, the plant types of the mutant and the wild type are measured, it mainly records the indexes of three-ear leaves, and the average value of each index of three-ear leaves represents the indexes of the whole plant. Compared with the wild type, the leaf length and height point length of the mutant *lpa1* are shorter, and the leaf width is narrower, but the height difference is significant; the leaf angle of the mutant *lpa1* is 38.94° while the leaf angle of the wild type is 33.19°, significantly increased; the leaf spacing of the mutant *lpa1* is 17.06 cm, while that of the wild type is 12.32 cm, and the leaf orientation value of the mutant *lpa1* is significantly smaller than that of the wild type (Table 1).

### 2.3. Analysis of Agronomic Characters of Maize Mutant lpa1

The survey results of agronomic characteristics of wild-type B73 and mutant *lpa1* are shown in Table 2. The plant height of mutant *lpa1* is 222.66 cm, wild-type B73 is reduced by 5.56%, and the plant height is significantly shorter; the ear position of the mutant is significantly lower than wild-type B73, the bottom was reduced by 18.99%; the number of tassel branches, the total tassel length, the tassel branch length, the panicle length and the panicle thickness of the mutant were compared with the wild type, except that there was no significant difference in panicle thickness. The mutants were significantly reduced by 53.57%, 24.25%, 40.61% and 26.17%, respectively. The single-ear weight, ear weight, seed setting rate and 100-kernel weight of the mutant were significantly lower than those of the wild type, and they were reduced by 54.13%, 60.21%, 13.26% and 34.19%, respectively. The various agronomic traits of the mutant and the wild type varied greatly.

### 2.4. Screening of the Optimal Concentration of Exogenous Hormones

The leaf angle is affected by many factors and is more sensitive to external signals. Both light and plant hormones may change the leaf angle. Current studies believe that the leaf angle is mainly related to plant hormones, and auxin plays an important role in the formation of leaf angles. Therefore, an auxin response analysis experiment was carried out on wild-type B73 and mutant *lpa1*. Two materials of three leaves and one heart stage were treated with different concentrations of hormones to analyze the effects of different concentrations of hormones on the size of the leaf angle. As the concentration of exogenous IAA increased, the leaf angles of wild-type and mutant became larger; when the concentration of IAA increased to 100 μmol·L^−1^, the wild-type leaves’ angles continued to increase, while the mutant’s gradually decreased. This indicates that *lpa1* is less sensitive to IAA (Figure 4A). Under the treatment of 100 μmol·L^−1^ IAA, the leaf angle of wild-type B73 was 39.6°, and the leaf angle of mutant *lpa1* was 60.6°. Source 100 μmol·L^−1^ IAA has the best effect on *lpa1*. Under the treatment of 500 μmol·L^−1^ IAA, the leaf angle of wild-type B73 was 50.3°, and the leaf angle of mutant lpa1 was 44.9°, which reached the maximum, and the difference between wild type and mutant was 5.4°. It can be concluded that the exogenous 100 μmol·L^−1^ IAA has the best effect on the wild type (Figure 4B).

### 2.5. Upset Plot Analysis of DEGs in B73 and lpa1 under Different Concentrations of IAA

To study the change of leaf angle between wild-type B73 and mutant *lpa1* under different concentrations of exogenous IAA treatment, we determined the DEGs under these conditions ((|Fold-change in expression| >1 and *p*-value < 0.05). Using FPKM as a measure of gene expression, a total of 10,958 and 7090 were identified in wild-type B73 and mutant *lpa1* under two concentrations of 100 μmol·L^−1^ IAA and 500 μmol·L^−1^ IAA DEGs (Figure 5). The Upset plot reflects the distribution of upregulation and downregulation of DEG in wild-type B73 and mutant *lpa1* under IAA treatment compared with CK treatment. At 100 μmol·L^−1^ under IAA treatment, compared with CK, 7522 DEGs (1733 upregulated and 5789 downregulated) were identified in wild-type B73 (Appendix A), and 3269 DEGs were identified in mutant *lpa1* (364 upregulated and 2905) downregulated (Appendix A). Compared with CK, 7700 DEGs (2529 upregulated and 5171 downregulated) were identified in wild-type B73 under 500 μmol·L-1 IAA treatment (Appendix A), and in the mutant *lpa1*, 5831 DEGs were identified (1766 upregulated and 4065 downregulated) (Appendix A). In the comparison of B73-CK vs. B73-100 and B73-CK vs. B73-500, a total of 2273 DEGs were identified. In the comparison of *lpa1*-CK vs. *lpa1*-100 and *lpa1*-CK vs. *lpa1*-500, 1017 DEGs were identified. At two IAA-treatment concentrations, a total of 516 DEGs were identified in wild-type B73 and mutant *lpa1*. These shared DEGs may play an important role in the adjustment of leaf angle in response to exogenous IAA in maize, reflecting the difference between the mutant *lpa1* and wild-type B73 has a difference in response to exogenous IAA.

### 2.6. Gene-Ontology Classification of DEGs in the Leaf Angle between B73 and lpa1 under Different IAA Concentrations

We used the gene-ontology (GO) function annotation to analyze all DEGs and divide them into biological processes, cellular components and molecular functions (Figure 6). Under 100 μmol·L^−1^ IAA treatment, the GO terms of DEGs in wild-type B73 were mainly annotated as cellular-component organization or biogenesis, cell-wall organization or biogenesis and cellulose synthase activity, and the GO terms of DEGs in mutant lap1 were mainly annotated as cellular-component organization or biogenesis, cell-wall organization or biogenesis and external encapsulating-structure organization. Under the treatment of 500 μmol·L^−1^ IAA, the GO terms of DEGs in wild-type B73 mainly annotated catalytic activity, carbohydrate metabolic process, and intrinsic components of membrane. The GO terms of DEGs in mutant lap1 were mainly annotated as cellular-component organization or biogenesis, cell-wall organization or biogenesis and external encapsulating-structure organization. This indicates that under the condition of exogenous IAA treatment, maize can regulate the change in leaf angle by regulating its cellular-component organization, cellular-component biogenesis, cell-wall organization and cell-wall biogenesis mechanism.

### 2.7. Pathway Enrichment Analysis of B73 and lpa1 at Different IAA Concentrations

We performed pathway enrichment analysis on all DEGs in wild-type B73 and mutant *lpa1*, and selected the top 20 metabolic pathways from each comparison (Figure 7). The DEG identified from the comparison of B73-CK vs. B73-100 and *lpa*1-CK vs. *lpa*1-100 is commonly enriched in indole alkaloid biosynthesis, starch and sucrose metabolism, diterpenoid biosynthesis, and plant–pathogen interaction pathways. The DEG identified from the comparison of B73-CK vs. B73-500 and *lpa*1-CK vs. *lpa*1-500 is commonly enriched in plant–pathogen interaction, starch and sucrose metabolism, indole alkaloid biosynthesis and flavone and flavonol biosynthesis pathways. Under IAA treatment, DEGs were enriched in metabolism, genetic information processing, organic systems, organic systems and environmental information processing of wild-type B73 and mutant *lpa*1. In the comparison of B73-CK vs. *lpa*1-CK, it was found that the common enrichment pathway of DEG and IAA-treated KEGG was indole alkaloid biosynthesis. We speculate that the indole alkaloid biosynthesis enrichment pathway related to the *ZmLPA*1 gene can regulate the size of the maize leaf angle.

### 2.8. Transcription-Factor Family Analysis

Transcription factors, also known as trans-acting factors, regulate gene expression at the transcriptional level and are proteins with special structures that can regulate plant growth and development. RNA-seq results showed that many TFs were differentially regulated under IAA treatment. Here, we only show TF families with more than three DEGs (Figure 8, Appendix A). We found that 30 families of TF were differentially regulated in wild-type B73 and mutant lap1 under IAA treatment. The top five TF families of DEGs are AP2-EREBP, bHLH, MYB, NAC and WRKY. The AP2-EREBP transcription factor is associated with plant-hormone signal transduction and MAPK signaling, the bHLH transcription factor is associated with plant-hormone signal transduction and MAPK signaling, the MYB transcription factor is associated with protein processing in the endoplasmic reticulum, and the WRKY transcription factor associated with the MAPK signaling pathway is related; we speculate that under the influence of exogenous IAA, maize can regulate the leaf angle through plant-hormone signal transduction, MAPK signaling and protein processing in endoplasmic reticulum-related transcription factors.

### 2.9. Validation of DEGs by qRT-PCR Analysis

In order to evaluate the reliability of the gene-expression profile of wild-type B73 and mutant *lpa1* in response to exogenous IAA under normal conditions and after the addition of different concentrations of exogenous IAA, qRT-PCR was used to verify the reliability of the gene-expression profile at 100 μmol·L^−1^ IAA and 500 μmol·L^−1^. The eight genes Zm00001d022388, Zm00001d026645, Zm00001d029366, Zm00001d042695, Zm00001d011086, Zm00001d053376, Zm00001d034298 and Zm00001d028662 were up-regulated in wild-type B73 and mutant *lpa1* at two concentrations of IAA (Figure 9). We found that the expression levels of wild-type B73 and mutant *lpa1* treated with different IAA concentrations of these eight genes are consistent with the transcriptome-sequencing results, indicating that our RNA-seq data is reliable.

## 3. Discussion

### 3.1. Functional Study of ZmLPA1 in Maize

Using ClustalW software to perform multiple alignments of ZmLPA1 amino acid se-quences, it was found that the ZmLPA1 gene in maize encodes a zinc-finger protein. which contains a plant-specific IDD domain, and the protein of this gene is named ZmIDD16/SGR5. The protein is connected to four different zinc-finger protein domains at the N-terminus, namely ZF1, ZF2, ZF3 and ZF4, which have conserved nuclear localization signals. These results indicate that ZmLPA1 belongs to the C2H2 transcription-factor-gene family. Using ProtComp v.9.0 to predict the subcellular localization of the ZmLPA1 gene, the results show that ZmLPA1 protein is localized in the nucleus, which is similar to the results of Wu et al. [26]. Bioinformatics analysis found that there are at least 20 *ZmIDD* genes in maize, at least 15 *OsIDD* genes in rice, and at least 16 *AtIDD* genes in Arabidopsis [5]. In Arabidopsis, *AtIDD3/MAGPIE*, *AtIDD10/JACKDAW* and *AtIDD8/NUTCRACKER* are involved in root development [28]. *AtIDD8* also regulates photoperiod flowering by regulating sucrose transport and metabolism [29]. *AtIDD1/ENHYDROUS* promotes seed germination by regulating the light and hormone signals during seed maturation [30]. *AtIDD5* actively regulates starch synthesis and regulates flower development. *AtIDD4* and *AtIDD6* regulate the synthesis of gibberellin. *AtIDD15/SGR5* is mainly expressed in the endothelium of the inflorescence stem and participates in the regulation of the early gravity response of Arabidopsis inflorescence [31]. In rice, three independent studies have shown that *OsID1* plays a decisive role in the flower development of rice [32,33]. Studies have found that the *OsIDD14/LPA1* is a functional homologous gene of *AtIDD15/SGR5* in Arabidopsis [26].

### 3.2. Plant-Type Characteristics of Mutant lpa1

In this study, a mutant *lpa*1 with increased leaf angle was selected from the wild-type B73 maize mutant library mutagenized by EMS. The plant height of the mutant *lpa*1 was significantly lower than that of the wild-type B73. From the great bell-mouth stage, the leaf angle was significantly different and the leaf angle of the mutant was larger than that of the wild type. The main stem and leaf ears of the mutant were red, the number of tassel branches was significantly reduced, the pollen vigor was weak, the mature ears of the mutants became smaller and the grains were oblate and smaller. Compared with the wild type, the leaf length and height point length of the mutant were shorter and the leaf width was narrower, but the difference in height point length was significant. The leaf spacing of the mutant was significantly longer and the leaf direction value was significantly smaller than that of the wild type. The survey results of agronomic traits showed that the mutant plant height, ear height, tassel branch number, tassel principal axis length, tassel branch length, ear length, single-ear weight, ear grain weight, seed setting rate and 100-kernel weight were significantly reduced, and there was no significant difference in ear thickness. The study found that the mutant *drl*1 exhibited leaf drooping. The mutation of the *drl1* gene affected the length, width and angle of the leaf. The study showed that the *drl1* gene controls the development of key agronomic traits in maize [25]. The mutant *ila*1 with increased leaf angle was obtained by T-DNA. The increase in leaf angle of *ila*1 was caused by the decrease in the mechanical strength of the leaf occipital part [34]. Studying the leaf photosynthetic characteristics of the ear position found that the height of the ear position of maize hybrids has the greatest influence on the grain weight per ear, followed by traits such as leaf length, leaf size and photosynthetic efficiency [35].

### 3.3. The Effect of Exogenous IAA on Maize Leaf Angle

Leaf occipital is a key factor that affects the formation of the leaf angle. Changing the cell division and cell elongation of the leaf occipital part will result in a change in the size of the leaf angle. Both external factors and internal growth regulators (such as hormones, cell-cycle regulators, cell-wall-modifying genes, etc.) can ultimately affect the size of the leaf angle by regulating cell division and cell elongation [36]. The main purpose of this study is to explore the regulation mechanism of plant hormones on maize mutant *lpa1* leaf angle, treat maize wild-type B73 and mutant *lpa1* with exogenous IAA and clarify that IAA regulates the development of leaf angles by controlling cell elongation or cell division in the occipital region of the leaf. The optimal hormone-concentration screening results showed that the leaf angle of the mutant *lpa1* increased first and then decreased with the increase in IAA concentration, and the leaf angle was the largest under the treatment of 100 μmol·L^−1^ IAA. The leaf angle of wild-type B73 increases with the increase in IAA concentration. Based on the results of cytological analysis, it is speculated that high concentrations of IAA may inhibit cell elongation and cell division in the occipital region of the mutant, and the appropriate concentration of IAA may increase the leaf angle of maize by regulating cell elongation and cell division in the occipital region.

### 3.4. Differentially Expressed Genes Comparison and GO Function Analysis

Through the statistics and comparison of the number of differentially expressed genes between different comparison groups, it was found that under exogenous hormone treatment, the number of differential genes in the wild type was more than that in mutants, and under exogenous hormone treatment, compared with the control, the number of downregulated genes in the material is greater than the number of upregulated genes, which is similar to the results of Liu et al. [27]. This study found that under the same concentration of exogenous IAA, the number of differentially expressed genes between the materials was higher, which was directly related to the mutation of *ZmLPA1*. GO annotation analysis of differential genes in different comparison groups showed that the differentially expressed genes in different comparison groups of the two materials were mainly enriched in the three biological processes of cellular processes, metabolic processes, and biological regulation, and were mainly enriched in cells and membranes. Among the four-cell components, membrane part and organelle, the number of corresponding differential genes is quite different in the molecular functions that are mainly enriched in catalytic activity and binding. Overall, by comparing the comparison group of mutant *lpa1* and wild-type B73 materials under the same treatment, it is found that the number of differentially expressed genes enriched in the above process under the exogenous IAA treatment is higher than the number of differentially expressed genes under the control condition, indicating that exogenous IAA can change the expression pattern of most genes, thereby regulating the size of the leaf angle.

### 3.5. Analysis of Differential Genes Related to Maize Leaf Angle under Different Concentrations of Exogenous IAA Treatment

This study is based on the results of pathway enrichment analysis and the series test of Cluster transcriptome sequencing. Seven candidate genes related to leaf angle were screened under the conditions of |log2FC| > 1 and *p* < 0.001; the study found that the expression of replication licensing factor MCM7 homologue9 (gi|542504|) was downregulated under the treatment of exogenous 100 μmol·L^−1^ IAA, but not under the treatment of exogenous 500 μmol·L^−1^ IAA, which indicates that the gene is under low-concentration IAA treatment and its expression level is downregulated, thereby promoting the growth and elongation of maize leaf cells, which may lead to an increase in leaf angle. Three genes are related to leaf growth and development. Plant-specific domain TIGR01627 family (gi|100282024|) was downregulated under the treatment of exogenous 500 μmol·L^−1^ IAA. It is a transcription factor that regulates plant growth and development. The connection is not yet clear, and further research is needed. Fasciclin-like arabinogalactan protein 11 (gi|103643069|) was downregulated under the treatment of exogenous 100 μmol·L^−1^ IAA, but not under the treatment of exogenous 500 μmol·L^−1^ IAA. It may be that the gene is more sensitive to IAA. It is expressed only when treated with an appropriate concentration of IAA, which mainly regulates the integrity of Arabidopsis cell wall, root growth and stem development, which may be related to leaf angle. Auxin response factor 15 (gi|103630727|) was downregulated under the treatment of exogenous 500 μmol·L^−1^ IAA, indicating that high concentrations of IAA inhibited the expression of this gene. It is speculated that auxin response factor 15 is positively correlated in auxin response. This study found three genes related to the auxin signal-transduction pathway, Auxin response factor 8 (gi|100194200|) was treated with 100 μmol·L^−1^ IAA. The expression was downregulated. The overexpression of this gene in Arabidopsis affects the elongation of the hypocotyl and the growth habit of the root system, indicating that it may participate in the dynamic balance of auxin under light conditions. Auxin response factor 4 (gi|100383226|) was downregulated under the treatment of exogenous 100 μmol·L^−1^ IAA. This gene regulates the growth of seedlings under light conditions by regulating auxin biosynthesis and auxin signal transduction and participates in it. The shade-avoidance-syndrome reaction may affect the size of the leaf angle, which is similar to the results of Hornitschek et al. [37]. The expression of Auxin response factor 16 (gi|100280136|) was downregulated under the treatment of exogenous 500 μmol·L^−1^ IAA, indicating that high concentrations of IAA inhibited the expression of this gene, weakened the cell division ability, and ultimately reduced the leaf angle. How the new genes discovered in the study respond to exogenous hormones to regulate the developmental mechanism of leaf angles remains to be further studied.

### 3.6. The Effect of ZmLPA1 on Maize Auxin Signal-Transduction-Related Genes

The development of the leaf angle is regulated by a variety of factors, and plant hormones play an important role in regulating leaf-angle development—especially auxin, which is very important for the regulation of leaf angle. AUX is the earliest discovered hormone with polar-transport characteristics. IAA is the main form of auxin, which can maintain the apical advantage of plants, promote cell elongation, root geotropism, embryo and endosperm development, etc. [12,38]. Auxin starts with tryptophan as a substrate and is synthesized through multiple pathways, the most important of which is the indole-3-pyruvate (IPA) synthesis pathway [39], It is mainly responsible for tryptophan aminotransferase TAA1 and flavin monooxygenase YUCCA [40,41]. We analyzed the pathway enrichment analysis of B73 and *lpa1* at different IAA concentrations and found that the common enrichment pathway of KEGG is indole alkaloid biosynthesis; in the process of *ZmLPA1* expression pattern, we found that its gene expression pattern is positively correlated with the concentration of AUX, we speculate that ZmLPA1 in plants may be related to the synthesis and transportation of AUX. Among the differentially expressed genes between mutant *lpa1* and wild-type B73, we found that the four types of genes that regulate AUX1, AUX/IAA, ARF, and SAUR, were significantly upregulated. The AUX1 cotransporter is responsible for the influx transport of auxin from extracellular to intracellular [42]; AUX/IAA and auxin response factor (ARF) form a dimer, thereby inhibiting the transcriptional regulatory function of ARF exercise. ARF-type transcription factors can bind to the cis-acting elements of auxin-induced genes to regulate their transcriptional expression [43]. Small auxin-up RNA (SAUR) is a fast-response gene for auxin, among which SAUR19 and SAUR63 have been found to be positive regulators of cell growth [44]. We speculate that *ZmLPA1* is related to the synthesis of maize AUX, and the mutant *lpa1* can respond more positively to exogenous IAA, increase the cell volume of the ears of maize and increase the angle of maize leaves (Figure 10).

## 4. Materials and Methods

### 4.1. Plant Materials and Planting Conditions

Mutant *lpa1* (EMS3-003c97) related to the expression of maize leaf angle was obtained from the maize EMS mutant library of Qilu Normal University (http://elabcaas.cn/memd/, accessed on 17 April 2021). The mutant *lpa1* was planted in the experimental field of Gansu Agricultural University; we used mutant *lpa1* to continuously backcross B73 for 2 generations and then self-brought for one generation to clear the mutant background, and we used the Sanger Method to preserve the mutated sites during background removal and observe mutant phenotypes. B73 germplasm material was provided by the Maize Breeding Research Group of the Agricultural College of Gansu Agricultural University. The experiment was carried out at the Gansu Provincial Key Laboratory of Aridland Crop Science, Gansu Agricultural University.

### 4.2. Mutant Phenotype Identification

We observed the morphology of the mutant and wild-type plants, and took pictures of the parts with obvious differences. During the pollination period, we measured the plant height, ear height and tassel-branch length of the mutant and wild type; measured the leaf length, leaf width, leaf angle, height point length and leaf spacing of three clover leaves during the pollination period; and measured the number of tassel branches. Agronomic traits such as ear length, ear row number, row grain number, single-ear weight, seed setting rate and 100-seed weight were determined.

### 4.3. Data Analysis

All data were analyzed using Microsoft Office 2019 and SPSS19.0 statistical software, and Duncan’s was used to analyze significant differences between the treatments, with three replicates for each treatment.

The gene sequence and amino-acid sequence were obtained using the NCBI (https://www.ncbi.nlm.nih.gov, accessed on 11 March 2021) database, and the subcellular localization was performed using the online program ProtComp v.9.0 (http://linux1.softberry.com, accessed on 11 January 2022). For prediction, ClustalW (https://www.ebi.ac.uk/Tools/msa/clustalw2, accessed on 12 January 2022) performed multiple alignments of amino-acid sequences; phylogenetic trees were constructed using MEGA7.0 software.

A series of quality-control steps were performed on the raw reads obtained by sequencing. The following reads were removed: reads containing adaptor sequences; reads with an unknown base number greater than 5% of the total sequence length; reads with Q ≤ 15; and reads where the base number accounts for more than 20% of the total read base number of the reads. Then, the clean reads for each sample were aligned to the third version of the B73 maize reference genome (http://ftp.maizesequence.org, accessed on 15 October 2021) using HISAT software [45]. Gene expression levels were determined using RSEM software [46]. DEGs were identified as described by Audic et al. [47]. DEG was defined as *p*-value < 0.05 and false discovery rate ≤ 0.001 compared to CK > 1 or <−1. The functional annotations of the selected DEGs in the GO database were determined using WEGO software [48]. The metabolic pathways enriched in DEGs were determined using the Kyoto Encyclopedia of Genes and Genomes pathway database (https://www.genome.jp/kegg/kegg1.html, accessed on 15 January 2022).

### 4.4. Extraction of Total RNA and Synthesis of First Strand cDNA

The total RNA from the leaf occipital part of maize B73 at the V3 stage was extracted, and the extraction procedure was carried out according to the instructions of Pure Plant Kit (DP423) of TIANGEN Biotech (Beijing, China) Co., Ltd. RNAprep (Beijing, China). The synthesis of the first strand of cDNA was carried out according to the instruction of FastKing RT Kit (With gDNase) (KR116) of Biotech (Beijing, China) Co., Ltd.

### 4.5. PCR Amplification of ZmLPA1 and Sequencing of the Product

Using Primer Premier 5.0 software, the primer sequences of *ZmLPA 1*(GRMZM2G465595) were: F-GGCCAGGACTCCCTTAAAAGAA, R-ATGAACTATGCATGGCAATGGC. Using wild-type B73 cDNA as a template, PCR amplification was carried out with the Easy Taq enzyme. The reaction system (20 μL) was as follows: Easy Taq 10 μL, ddH2O 7 μL, forward and reverse primers 1 μL each, and cDNA template 1 μL. Amplification conditions were as follows: pre-denaturation at 94 °C for 4 min, 94 °C for 1 min, 56 °C for 30 s, 72 °C for 1 min; after 35 cycles, extension at 72 °C for 10 min. We sent the PCR-amplification products to Xi’an Tsingke Company for sequencing.

### 4.6. Bioinformatics Analysis of ZmLPA1

We used the NCBI database to find the gene sequence and amino-acid sequence encoded by ZmLPA1; DNAMAN software to analyze *ZmLPA1* sequence and sequence alignment; ClustalW program to perform multiple alignments of amino acid sequences; and MEGA7.0 software to construct multiple alignment results generated by ClustalW Phylogenetic tree.

### 4.7. Material Hormone Treatment

We used vermiculite to plant maize seeds in a flowerpot with a diameter of 10 cm. The flowerpot was placed in a light incubator (28 °C, 65% RH, 14 h, 8000 Lux, 25 °C, 65% RH, 10 h, 0 Lux), When the maize seedlings grew to V3 stage, we sprayed different concentrations of exogenous IAA (0, 0.5, 1, 10, 100, 300, and 500 μmol·L^−1^), then cultivated the maize seedlings for 12 h, and sprayed again with different concentrations of exogenous IAA; 24 h after the processing is completed, the size of the leaf angle under different processing concentrations was counted, and the material of 2 cm at the leaf occipital area was taken and sent to the company for sequencing.

### 4.8. Total RNA Extraction and Detection

TRIzol reagent (Invitrogen) was used to extract total RNA from the leaf occipital part samples (a total of 18) of *lpa*1 and B73 plants under different conditions. To ensure the reliability and comprehensiveness of the sequencing data, the Agilent 2100 Bioanalyzer and NanoDrop were used to detect the total RNA concentration and quality of the extracted samples before sequencing. Using the BGISEQ-500 sequencing platform for RNA-Seq sequencing, the average number of clean databases produced by each sample is 11.07Gb; The average output was 90,910,840 original reads, and the quality control of the original reads yielded clean reads of maize leaves. The average number was 77,386,306. The size of the filtered data accounted for 85.13% of the original data size; the percentage of bases with a quality value of Q ≥ 30 was 84.88% and above. Comparison of clean reads was obtained from the screen to the B73 RefGen v4 maize reference genome sequence using HISAT software. The original sequencing reads were submitted to the SRA at NCBI (Accession number: PRJNA821235).

## 5. Conclusions

In this study, the phenotype and genotype of mutant *lpa1* and wild-type B73 were identified from three aspects: agronomic traits, gene identification and bioinformatics analysis. Different concentrations of exogenous IAA were used to screen the V3 phase mutant *lpa1* and wild-type B73, and two optimal concentrations were obtained from hormone-response levels, and the function of *ZmLPA1* was analyzed by transcriptome-sequencing results. The results showed that the expression pattern of the *ZmLPA1* gene was correlated with the distribution of auxin in maize plants. Among the KEGG enrichment pathways, we found a common enrichment pathway for indole alkaloid biosynthesis, and found seven candidate genes related to leaf angle from the differentially expressed genes. From the analysis of our obtained results, *ZmLPA1* is involved in the synthesis and response of AUX in maize plants, and *ZmLPA1* regulates maize leaf angle by regulating maize auxin. Our research results have important theoretical significance for analyzing the molecular mechanism regulating leaf angle of maize and provide a promising new gene resource for shaping the ideal plant type.

## Figures and Tables

**Figure 1 ijms-23-04886-f001:**
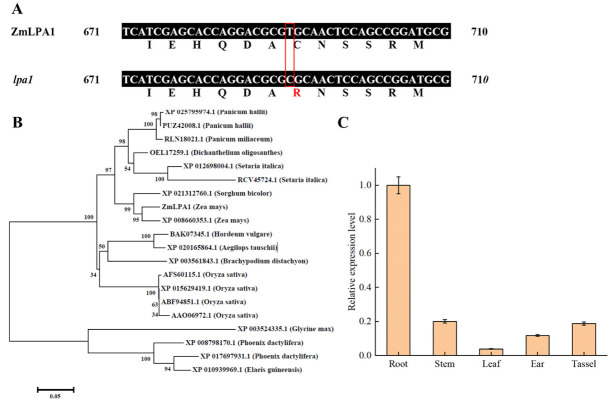
Analysis of *ZmLPA1* gene in maize. (**A**) Alignment of *ZmLPA1* gene cDNA sequences in B73 and *lpa1*, amino acid changes are marked with red box; (**B**) Phylogenetic tree of *ZmLPA1* gene; (**C**) Transcription expression of *ZmLPA1* gene in different tissues of maize.

**Figure 2 ijms-23-04886-f002:**
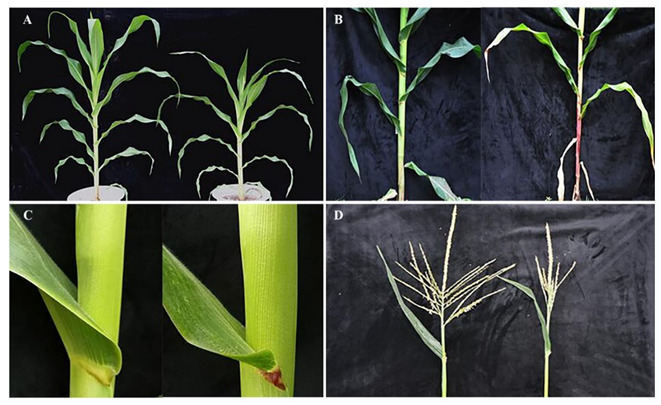
Phenotypic analysis of wild-type and mutant *lpa1*. (**A**) Plants at trumpet stage; (**B**) main stem at male drawing stage; (**C**) corresponding leaf angle in A; (**D**) male spike: the left side is wild type and the right side is mutant.

**Figure 3 ijms-23-04886-f003:**
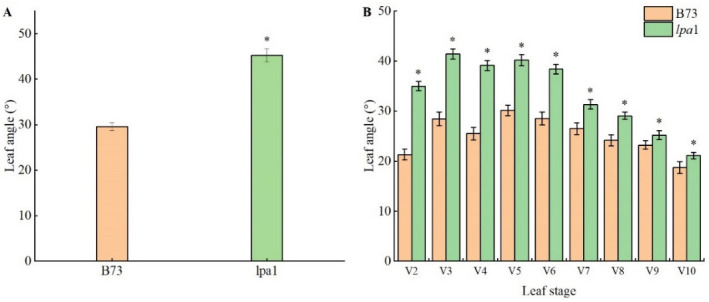
Analysis of index of wild-type and mutant *lpa1*. (**A**) Leaf angle with the greatest difference; (**B**) leaf angle of 2–10 leaves in trumpet stage. * indicates *p* < 0.05.

**Figure 4 ijms-23-04886-f004:**
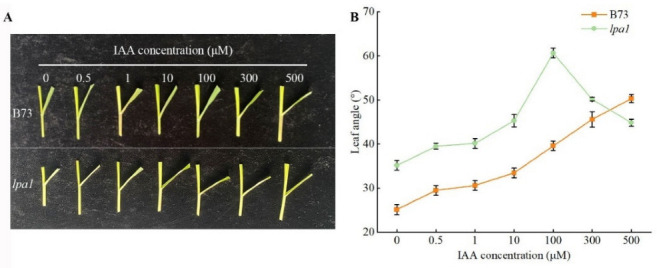
Effects of exogenous hormones on leaf angle of wild-type B73 and mutant *lpa1*. (**A**) Effects of different concentrations of IAA on leaf angle of wild type and mutant. (**B**) Size of leaf angle for wild type and mutant under different concentrations of IAA treatment.

**Figure 5 ijms-23-04886-f005:**
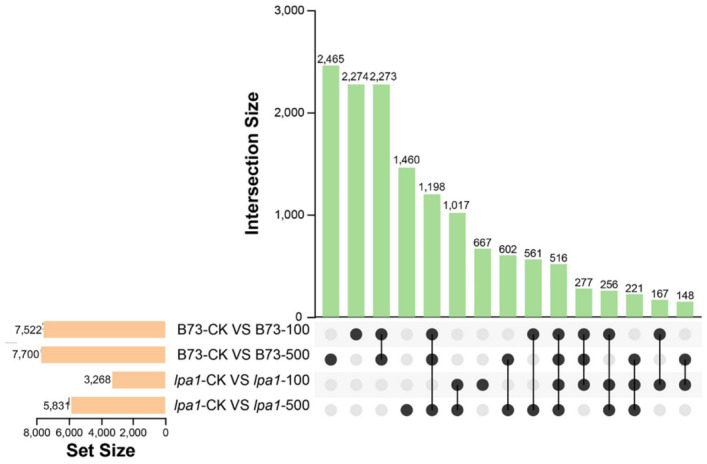
Upset plot analysis of DEGs in B73 and *lpa1* under different treatments. The columns indicate DEGs under single or multiple processes, the *x*-axis represents combinations of different groups (black dots represent single-treatment comparisons, black lines connected by dots represent multiple-treatment comparisons) and the *y*-axis represents the number of genes corresponding to the combinations.

**Figure 6 ijms-23-04886-f006:**
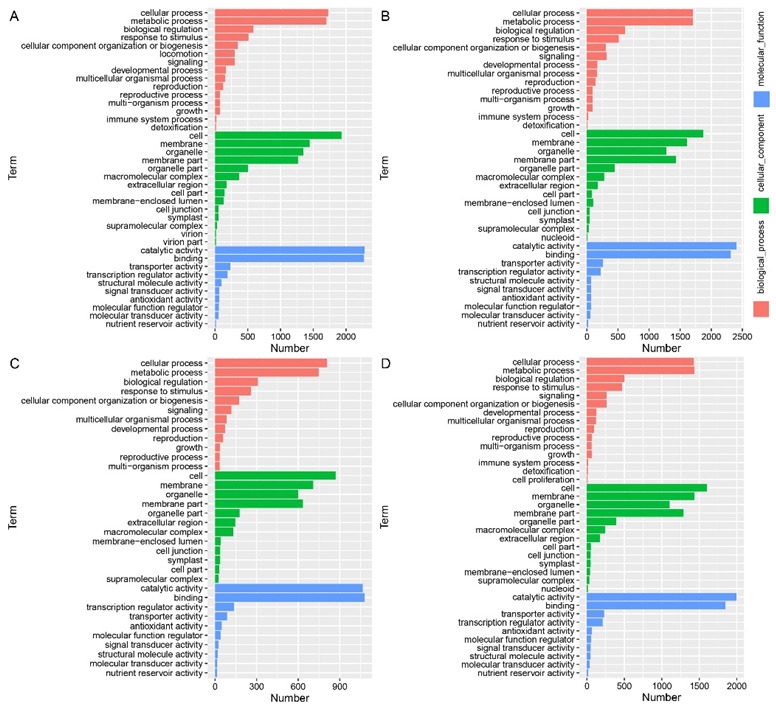
GO annotations of DEGs in B73 and *lpa1* identified under different treatments. (**A**) B73-CK vs. B73-100; (**B**) B73-CK vs. B73-500; (**C**) lpa1-CK vs. lpa1-100; (**D**) lpa1-CK vs. lpa1-500. CK = control, 100 = 100 μmol·L^−1^ IAA, and 500 = 500 μmol·L^−1^ IAA.

**Figure 7 ijms-23-04886-f007:**
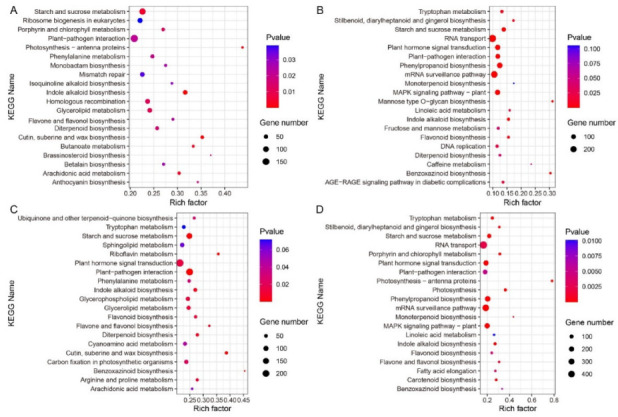
Pathway enrichment analysis of B73 and *lpa1* exposed to different treatments. (**A**) B73-CK vs. B73-100; (**B**) B73-CK vs. B73-500; (**C**) *lpa1*-CK vs. *lpa1*-100; (**D**) *lpa1*-CK vs. *lpa1*-500. CK = control, 100 = 100 μmol·L^−1^ IAA, and 500 = 500 μmol·L^−1^ IAA.

**Figure 8 ijms-23-04886-f008:**
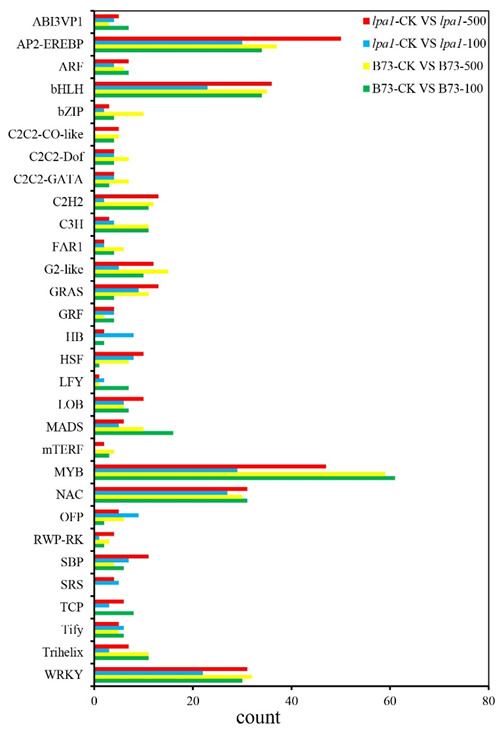
Analysis of transcription factors differentially expressed in B73 and *lpa1* under different treatments.

**Figure 9 ijms-23-04886-f009:**
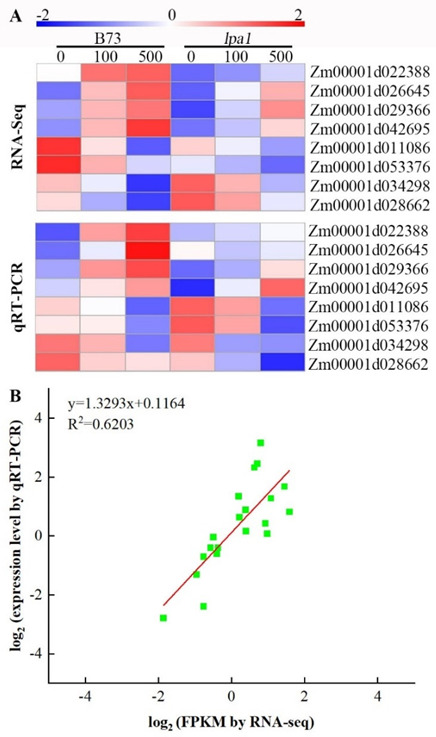
The expression pattern of eight selected genes identified by RNA-seq was verified by qRT-PCR. (**A**) Heat map showing the expression changes (log_2_ fold change) in response to the B73-CK to *lpa1*-500 treatments for each candidate gene as measured by RNA-seq and qRT-PCR; (**B**) Scatter plot showing the changes in the expression (log-fold change) of selected genes based on RNA-seq via qRT-PCR. The gene expression levels are indicated by colored bars.

**Figure 10 ijms-23-04886-f010:**
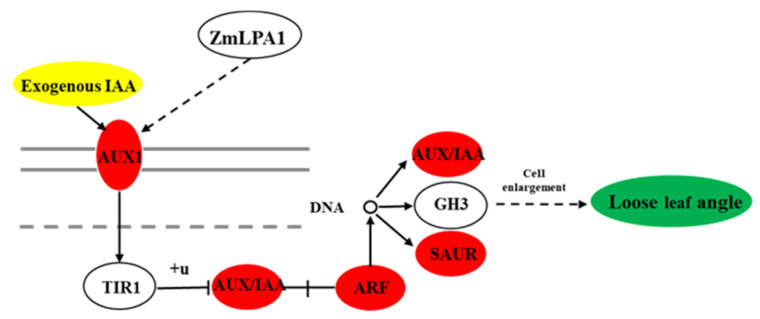
*ZmLPA1* gene regulates maize leaf angle. *ZmLPA1* acts as a central gene that directly regulates the auxin signaling pathway, thereby regulating the size of leaf angle. The red circle indicates that the expression of auxin-related genes in mutant *lpa1* is higher than that in wild-type B73 under exogenous IAA treatment.

**Table 1 ijms-23-04886-t001:** Analysis of relevant characters of wild-type B73 and mutant *lpa1* cloverleaf.

Traits	B73	*lpa1*
Leaf length/cm	84.68 ± 6.86 a	81.43 ± 4.89 a
Leaf width/cm	14.16 ± 0.29 ab	10.20 ± 0.32 b
Leaf angle/°	33.19 ± 0.44 b	38.94 ± 3.53 a
High point length/cm	62.70 ± 4.70 b	55.49 ± 2.29 a
Leaf space/cm	12.32 ± 0.16 b	17.06 ± 0.58 a
Leaf orientation value	42.11 ± 1.64 b	34.78 ± 1.44 a
Plant height/cm	235.78 ± 0.92 a	222.66 ± 0.66 b

Note: Lowercase letters indicate a significant difference at 5% level.

**Table 2 ijms-23-04886-t002:** Investigation of agronomic characters of wild-type B73 and mutant *lpa1*.

Traits	B73	*lpa1*
Plant height/cm	235.78 ± 0.92 a	222.66 ± 0.66 b
Ear height/cm	131.18 ± 0.43 a	106.26 ± 0.32 b
Number of tassel branches	11.50 ± 0.41 a	5.50 ± 0.41 b
Total tassel length/cm	38.55 ± 0.04 a	29.20 ± 0.24 b
Tassel branch length/cm	27.95 ± 0.37 a	16.60 ± 0.08 b
Panicle length/cm	20.90 ± 0.33 a	15.43 ± 0.67 b
Per-panicle weight/g	213.83 ± 2.43 a	98.08 ± 1.27 b
Kernel weight/g	186.87 ± 1.28 a	74.35 ± 0.80 b
Seed setting rate/%	87.40 ± 0.39 a	75.81 ± 0.17 b
Hundred-grain weight/g	43.99 ± 0.73 a	28.95 ± 0.65 b
Ear coarse/cm	5.72 ± 0.02 a	4.38 ± 0.06 a

Note: Lowercase letters indicate a significant difference at 5% level.

## Data Availability

Not applicable.

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
