# Peer review of "Mutant lpa1 Analysis of ZmLPA1 Gene Regulates Maize Leaf-Angle Development through the Auxin Pathway"

_ijms, 2022, doi:10.3390/ijms23094886_

Round 1

Reviewer 1 Report

The study title is focused on Mutant lpa1 Analysis of ZmLPA1 Gene Regulates Maize Leaf 2 Angle Development Through the Auxin Pathway. Study is clearly described. Please add all limitations of the study.

Author Response

Dear reviewer:

We appreciate these valuable comments and suggestions very much. We have made detailed corrections in the manuscript corresponding to your suggestions and advice. Thank you for your time and consideration. According to the reviewers' comments, we have seriously revised those problems. At the same time, we have also made other changes.

Point 1:  The study title is focused on Mutant lpa1 Analysis of ZmLPA1 Gene Regulates Maize Leaf Angle Development Through the Auxin Pathway. Study is clearly described. Please add all limitations of the study.

Response 1: Thanks for your important comment and suggestion. We think your suggestion is very good. We have added missing information to the manuscript, revised and supplemented the imperfect sections of the manuscript, and marked the revised and supplemented sections in red font.

Thank you again for your detailed and significant suggestions, and hope that the corrections will meet with your approval.

Best wishes!

Reviewer 2 Report

The manuscript covers the phenotype of the lpa1 mutant of Zea mais. The phenotype and mutation in the mutant line were analysed, and the transcriptomes of the mutant and the wild-type plants were investigated.

Comments and suggestions to authors:

Introduction – not enough background information is given about the Lpa gene. Nothing is mentioned about the lpa1 line.

L83-88. Not clear, how was the mutant first found. Needs clarification.

L89-90 – “Mutations in the ZmLPA1 gene lead to changes in leaf angle in wild-type B73 maize.” Were there gene substitution experiments carried out? If not – the sentence is unclear.

Lines 90-91 Unclear which plants were compared

Figure 1 b Authors should consider changing accession numbers to something more human-readable, like species names

L 171 – consider rephrasing the name of this part. Additionally the upset plot and the Venn diagram contain the same information. It would be more informative if the data for upset were split the same way as for the Venn plots. This was it would be more clear which groups behave differently in the investigated lines. Venn pots could be removed then.

Figure 6 is nigh-unreadable. I would suggest leaving only the only the most important part, or alternatively splitting the figure in three and rotating the axis. Additionally only the broadest terms were used for annotation, leading to obvious groups, which do not tell anything about the underlying biology.

L237 Not enough information is given in this part to objectively judge the observations.

L273-276 Not clear how is this the result of this study. The domain structure was described previously in e.g. https://doi.org/10.1111/j.1365-313X.2006.02807.x

L277 – Can not be considered the result of transcriptome sequencing

L291-292 – Unclear why does this gene belong to a new subfamily

L307 – The process of choosing the 7 genes is unclear.

Figure 10 – Unclear where the LPA1 gene is in the figure

L 472 – not enough information is given bout the exact kit or process used to obtain the library, additionally no information is given about the software used to clean the reads. No information is given about the methods used for DGE calculation, the enrichment analysis or other analyses.

No information is given about the reads. The raw read data should be in an accessible database, NCBI of analogous. No information is given about the genome version used in this study.

The methods used in data analysis are not described in the text, making it impossible to judge the manuscript fairly. The necessary information should be added for the next review round.

Author Response

Dear reviewer:

We appreciate these valuable comments and suggestions very much. We have made detailed corrections in the manuscript corresponding to your suggestions and advice. Thank you for your time and consideration. According to the comments, we have studied the comments carefully and have made a correction which we hope meet with approval. At the same time, we have also made other changes. Please refer to the Response to Reviewers 2.

Thank you again for your detailed and significant suggestions. Based on your comments, we have revised the corresponding content in the manuscript and hope that the correction will meet with your approval.

Best wishes!
